# Epizoochory in Parrots as an Overlooked Yet Widespread Plant–Animal Mutualism

**DOI:** 10.3390/plants10040760

**Published:** 2021-04-13

**Authors:** Dailos Hernández-Brito, Pedro Romero-Vidal, Fernando Hiraldo, Guillermo Blanco, José A. Díaz-Luque, Jomar M. Barbosa, Craig T. Symes, Thomas H. White, Erica C. Pacífico, Esther Sebastián-González, Martina Carrete, José L. Tella

**Affiliations:** 1Department of Conservation Biology, Doñana Biological Station (CSIC), Calle Américo Vespucio, 26, 41092 Sevilla, Spain; hiraldo@ebd.csic.es (F.H.); jomarmbarbosa@gmail.com (J.M.B.); ericapacifico81@gmail.com (E.C.P.); 2Department of Physical, Chemical and Natural Systems, Universidad Pablo de Olavide, Carretera de Utrera, km 1, 41013 Sevilla, Spain; pedroromerovidal123@gmail.com (P.R.-V.); mcarrete@upo.es (M.C.); 3Department of Evolutionary Ecology, Museo Nacional de Ciencias Naturales (CSIC), Calle José Gutiérrez Abascal, 2, 28006 Madrid, Spain; gblanco@mncn.csic.es; 4Foundation for the Research and Conservation of Parrots in Bolivia (FPCILB), Avda. Mariscal Sta. Cruz, 5030 Santa Cruz de la Sierra, Bolivia; jdiaz@fclbolivia.org; 5School of Animal, Plant and Environmental Sciences, University of the Witwatersrand, Private Bag 3, Wits 2050, South Africa; craig.symes@wits.ac.za; 6US Fish and Wildlife Service—Puerto Rican Parrot Recovery Program, Box 1600, Rio Grande, PR 00745, USA; thomas_white@fws.gov; 7Department of Ecology, Universidad de Alicante, Carretera de San Vicente del Raspeig, s/n, 03690 Alicante, Spain; esebgo@gmail.com

**Keywords:** plant–animal mutualism, seed dispersal, zoochory, Psittaciformes, Moraceae, biological invasions, citizen science

## Abstract

Plant–animal interactions are key to sustaining whole communities and ecosystem function. However, their complexity may limit our understanding of the underlying mechanisms and the species involved. The ecological effects of epizoochory remain little known compared to other seed dispersal mechanisms given the few vectors identified. In addition, epizoochory is mostly considered non-mutualistic since dispersers do not obtain nutritional rewards. Here, we show a widespread but unknown mutualistic interaction between parrots and plants through epizoochory. Combining our observations with photos from web-sources, we recorded nearly 2000 epizoochory events in 48 countries across five continents, involving 116 parrot species and nearly 100 plant species from 35 families, including both native and non-native species. The viscid pulp of fleshy fruits and anemochorous structures facilitate the adherence of tiny seeds (mean 3.7 × 2.56 mm) on the surface of parrots while feeding, allowing the dispersion of these seeds over long distances (mean = 118.5 m). This parrot–plant mutualism could be important in ecosystem functioning across a wide diversity of environments, also facilitating the spread of exotic plants. Future studies should include parrots for a better understanding of plant dispersal processes and for developing effective conservation actions against habitat loss and biological invasions.

## 1. Introduction

The knowledge of biotic interactions is critical to understand the structure of communities and ecosystem function [1,2], as well as to develop effective conservation actions against increasing habitat loss worldwide [3,4]. These complex interaction networks, such as those that arise from plant–animal interactions, are key to the support of ecosystem dynamics and diversity of both animals and plants [1,5]. For example, frugivores increase germination and dispersal of their food plants due to their role as seed dispersers [6] and, consequently, not only can they influence the composition and abundance of plant communities but they can also trigger cascading effects on other communities [7,8,9]. However, the wide array and complexity of biotic interactions can make many of them remain unknown or not fully understood [10].

Among different seed dispersal mechanisms, epizoochory (i.e., the dispersal of animal and plant propagules adhered to the body surface of animals, [11]) is one of the zoochory syndromes less studied, as we have important knowledge gaps on its frequency, efficiency and the extent of its effects on plant communities [12,13,14,15]. Plants with epizoochory syndromes usually show specialized structures in their seeds such as hooks or barbs that facilitate the adherence and attachment of propagules [11,12,13,16], although there are examples of seeds that stick to animal body parts when mixed with mud [17]. Effective attachment mechanisms can assist long retention times on the disperser, increasing the likelihood to disperse over long distances. Once seeds adhere to the animal surface, their detachment can be facilitated by certain behaviors of the dispersers, such as grooming and bill-wiping [18,19] and conspecific social interactions, which could even assist secondary seed dispersal events after seed transference between individuals [13,20]. These seeds, in general, attach inadvertently to the animal’s body without a clear benefit to the transporter [11,13]. Sometimes, however, tiny seeds within the viscid pulp of some fleshy fruits can attach to animals and be transported without seed anchoring structures but glued by the residues of the fruit’s pulp [21]. This infrequent form of epizoochory, which is even less known, can be considered as mutualistic, contrary to the other types where only plants are benefitted. Previous studies have shown that epizoochory in certain groups such as terrestrial birds is anecdotic [21,22,23] and largely skewed to waterbirds, probably because plants exploited by terrestrial birds are rarely epizoochorous [21,23].

Here, we show that epizoochory is a widespread mutualistic dispersal mode among parrots (Order Psittaciformes), associated with fruit consumption. Parrots are a diverse group of fruit and seed generalists that exploit a high diversity of plants and plant parts in different maturation stages [24,25]. Widely considered as seed predators, the mutualistic roles of parrots in studies of plant–animal interactions have been mostly neglected [26,27]. Recent studies, however, have largely proved their role as legitimate primary seed dispersers (e.g., [25,28,29,30]) and promoters of secondary seed dispersal [31]. The small size of the seeds that could be transported by parrots through epizoochory [32,33], together with the often fast and hidden movements of parrots while feeding inside the vegetation, have probably made this an overlooked behavior so far. Therefore, our main aim was to obtain solid evidence that parrots can also play a seed dispersal role through epizoochory. To this end, we combined direct observations and photos to detect events of epizoochory between parrots and their food plants worldwide, providing also information on dispersal distances and traits of the seeds dispersed. Our study highlights how unexpected biotic interactions may be largely overlooked, providing thus useful information towards understanding ecosystem structure and functioning in the current scenario of global change.

## 2. Results

### 2.1. Epizoochory Events and Dispersal Behavior

A total of 1892 individual parrots carrying adhered seeds were recorded from our direct observations (90.5%) as well as web-sourced pictures (9.5%) taken by 158 wildlife photographers (Appendix A), showing how they may have inadvertently contributed to citizen science. Observations were obtained in 48 countries, including 10 where parrots are non-native (Figure 1). We recorded these interactions in a wide range of environments, including continents and islands, rain and dry forests, and natural and urban areas, involving 116 parrot species from 48 genera. Parrots ranged in size from the smallest (green-rumped parrotlet *Forpus passerinus*, 12 cm) to the largest species (scarlet macaw *Ara macao*, 85 cm). The propagules transported (Figure 2) were always from vascular plants except in one case (a lichen). We identified the species in 91% of cases (N = 1722), the genus in 95% of cases (N= 1795), and the family of plants in 96% of cases (N = 1812), corresponding thus at least to 96 species from 35 families. Moraceae was the main family involved, followed by Salicaceae and Amaranthaceae (Figure 3).

Nearly half of the parrot species (N = 50) interacted with more than one plant species (Figure 4, range: 1–28, mean = 2.38), thus resulting in 276 interactions between parrot and plant species. Most observations of non-native parrots (80.7%) involved exotic plant species, while this happened with only 8.6% of the observations involving native parrots. Observations of exotic parrots were mainly conducted in urbanized areas (70.1%), while native parrots in urban environments were 46.7% of the total. In most instances (65.63% of plant species), tiny seeds were attached to the beak or head feathers of individuals when they were feeding on the pulp of fleshy fruits (e.g., family Moraceae) (Figure 2a,b,d,f,g,h). In fewer cases (16.67% of plant species), cottony structures containing small seeds of anemochorous fruits (e.g., family Malvaceae) were attached to their beaks or feet (Figure 2i). In the rest of the cases, seeds came from dry dehiscent fruits (e.g., capsules and legumes) that were also attached to the facial feathers and the beak through resins and mucilages in seeds (Figure 2e,j). The dimensions of the seeds (i.e., length and width) attached and subsequently dispersed by parrots ranged from 0.7 (*Aizoon canariense*) to 12.5 mm (*Pseudopondias microcarpa*) (mean ± SD = 2.73 ± 1.38 mm) and 0.4 (*Patellifolia procumbens*) to 9.25 mm (*Protium heptaphyllum*) (mean ± SD = 1.59 ± 1.04 mm), respectively (Figure 5a). However, if only the measurements per dispersed plant species are considered, they showed larger mean sizes both in terms of length (mean ± SD = 3.70 ± 2.44 mm) and width (mean ± SD = 2.56 ± 1.79 mm) (Figure 5b). Nearly 90% of observations corresponded to seed sizes (both in length and width) smaller than or equal to the mean seed size of the plant species dispersed, indicating that most seeds dispersed were within the lower range distribution of seed size, probably because smaller seeds are more easily attached.

We observed that parrots usually dropped seeds that were detached from their body surface in grooming sessions with the beak or by rubbing them against tree branches. Conspecific interactions such as social grooming, courtship feeding, and parental care (Figure 2c) were also witnessed promoting the removal or transfer of seeds between individuals when attached seeds were dropped.

### 2.2. Seed Dispersal Distances

Seed dispersal distances were measured in 112 epizoochory events. Straight-line distances covered by 17 parrot species when they flew with attached seeds after foraging ranged from 12 to 452 m (mean = 118.5 m, median= 100 m, Figure 6). It is worth noting that these seed dispersal distances are minimum values as, in most cases, we lost the parrots while flying or could not confirm whether they dropped the seeds where they first perched or later on, after moving larger distances by flying until other perching sites.

## 3. Discussion

### 3.1. Epizoochory in Parrots as a Widespread Mutualism 

In this study, we found that epizoochory in parrots adds a new example of understudied animal-mediated dispersal mode [15] that challenges our understanding of this dispersal syndrome in several ways. First, epizoochory is generally considered infrequent compared to other zoochory mechanisms such as endozoochory (i.e., dispersal of viable seeds after gut passage), as it is mostly caused by the eventual attachment of propagules to the body of mammals or some bird species [11,16,17,34]. These plant–animal interactions are not considered mutualistic, as adhered fruits do not provide dispersal agents with a nutritional reward [11,16], with the only exception of some extremely rare events of epizoochory in songbirds [21,22]. Here, however, we are providing evidence of epizoochory as a taxonomically and geographically widespread mutualistic interaction, since parrots eat the pulp and/or seeds of the plants they are dispersing. Second, epizoochory in mammals occurs thanks to specialized fruit structures of some plants (barbs, hooks, or viscid outgrowths) that allow their attachment to fur [11,12], suggesting an epizoochorous dispersal syndrome for these plants. However, parrots do not disperse fruits through epizoochory but their seeds and the plants dispersed are traditionally considered to belong to endozoochorous or anemochorous (i.e., seeds dispersed by wind) dispersal syndromes [35,36]. Observations in parrots thus question the blind identification of dispersal syndromes by just attending to fruit morphology [11,30,37]. Third, contrary to other epizoochorous vectors, parrots may enhance seed viability since they remove (ingest) the pulp that often contains germination inhibitors [38,39] before the dispersal of the attached seeds. Fourth, plants typically dispersed by epizoochory do not attract animals and, thus, encounters with dispersers occur by chance, driving low fruit removal rates [11]. Contrarily, parrots are attracted by the fruits they eat and may transport adhered seeds very frequently. The attachment of seeds to the bill of other frugivorous or granivorous birds seems to be anecdotal [21,22], although the same limiting factors during observations (e.g., small seed sizes and cryptic behavior during foraging) could also overlook this seed dispersal mechanism in these groups as well as underestimate their effects. However, the unique fruit manipulation abilities of parrots, using their large and mobile beaks, their tongues, and their feet for feeding [25], may facilitate high rates of seed attachment through pulp residues of fleshy fruits, viscid mucilages, and seeds with specialized structures for anemochory that easily adhere on their body surfaces. Finally, coinciding with typical epizoochory syndromes [11,16], parrots may perform directed dispersal by moving seeds far from the mother plant to similar habitats where germination is feasible.

### 3.2. Ecological and Conservation Implications

Our preliminary data depict an overlooked plant–animal mutualism that merits further research. Although we found evidence of epizoochory in c. 30% of the extant parrots of the world, we expect that other researchers, after combining observations with picture-based citizen science, will substantially increase the number of species and pairwise interactions worldwide. Thus, other plant species with small seeds could be dispersed by parrots independently of their sizes, which contrasts with other seed dispersal mechanisms, such as endozoochory, in which the size of the dispersing animal is determinant [40,41]. More fieldwork and experiments are needed to ascertain the frequency of epizoochory, retention times of seeds, dispersal distances, and seed fate. Parrots conduct extensive movements within their foraging ranges [25], which would enhance epizoochory effectiveness [14]. Regarding retention times, we recorded in our study a breeding female rose-ringed parakeet (*Psittacula krameri*) that after feeding on black mulberry fruits (*Morus nigra*) held several seeds attached to its facial feathers for at least three days. Thus, retention times of adhered seeds could be potentially larger than those recorded in endozoochory [11,42], promoting long-distance dispersal. On the other hand, non-native parrots largely assist seed dispersal of exotic plants. This is explained because our records of exotic parrots were mainly obtained in urbanized environments, where the abundance and diversity of exotic taxa are higher than in natural environments [43,44]. Despite exotic plants show lower frequencies of seed dispersal by native parrots, plant invasions can be also facilitated by native frugivores [45]. Thus, the consumption of exotic plants by both native and exotic parrots in urban areas can trigger their spread across surrounding natural landscapes through seed dispersal [46,47,48]. Given that parrots are successful invaders worldwide [49], with nearly 16% of the extant parrot species showing non-native populations out of their natural geographical distribution [50], further research should assess whether their interactions with exotic plants can be promoting their spread. These potential interactions may trigger plant invasions, increase their impacts through invasional meltdown processes [51], and disrupt the existing animal-mediated dispersal systems [16,52], through the competence with native plants for the attention of potential seed dispersers [51,53]. Additionally, we have observed that some exotic plants and their exotic dispersers coexist in their native ranges. For instance, several fig species (genus *Ficus*) from southern Asia and Africa were introduced in European urban areas together with the rose-ringed parakeet. The same pattern is observed with pepper tree species (genus *Schinus*) and the monk parakeet (*Myiopsitta monachus*) introduced in Europe, both native to southern South America. These co-occurrences between species out of their native ranges keeping their mutualistic interactions can develop synergetic impacts and facilitate their invasion processes [51,54]. Finally, the complementarity between epizoochory and other dispersal mechanisms [21] also needs further research.

In recent years, parrots have been shown to play a dual role as seed predators and primary seed dispersers through endozoochory [30,32,55] and estomatochory (i.e., plant propagules are purposely transported with their beaks and dropped after fruit consumption, e.g., [9,28,29,56,57,58]), also facilitating secondary seed dispersal [31]. We have already identified some plant species dispersed by parrots through all these dispersal mechanisms (e.g., Figure 2j), so epizoochory (acting alone or together with others) should be further considered when assessing the role of parrots on ecosystem functioning [26] and network structure [27]. In a scenario of global change where unstoppable habitat loss continues, plants with zoochory mechanisms may be less vulnerable to perturbations [59,60] and these, together with their dispersers, provide key services for the recolonization and recovery of forested habitats [4,15,36]. Thus, we consider that advances in the knowledge of seed dispersal by parrots and future efforts for understanding additional biotic interactions will assist in the development of effective conservation actions against both habitat loss and biological invasions.

## 4. Materials and Methods

### 4.1. Data Recording and Source

To assess epizoochory in parrots, we mainly used information from our observations of seed dispersal in different fieldwork campaigns performed in 17 countries and 5 continents (Figure 1) between 2012 and 2020. We actively looked for foraging groups of parrots both through roadside surveys and walking transects across a variety of biomes and habitats [31]. We conducted 98 roadside surveys covering c. 57,250 km of transects [61], and a large-but unquantified-number of walking transects to look for foraging parrots. Once parrots were detected, we tried to observe individuals at a distance for 5–15 min and to record the potential presence of seeds attached to the surface of parrots (mainly beak or head) after fruit consumption. We were able to detect attached seeds by using binoculars, telescopes, and telephoto lens. We were able to identify the parrot species, also recording flock size, the ripening stage of the fruits/seeds (unripe/ripe), the attachment mechanism of seeds (fleshy, anemochorous, and others), date, and the location where the observation was recorded. Regarding plants consumed by parrots, we identified the species whenever possible. Observations taken out of the native range of a parrot or a plant species were classified as exotics.

To find more evidence of epizoochory in parrots, we carefully reviewed our photo gallery from our fieldwork campaigns. Besides, to check whether wildlife photographers might have also captured this phenomenon in their pictures, we made a non-exhaustive viewing of some photo galleries publicly available on the internet, namely: eBird, WikiAves, Flickr, Instagram and Facebook (Appendix A). Photographs were visually examined and strictly considered as epizoochory events when undamaged seeds attached on parrots were discernible from other residues of consumed plants, such as the pulp. Plant species were identified when possible. Otherwise, they were categorized as “Unidentified” or “Unknown” if they were unidentifiable to the genus- and/or family level, respectively (Appendix A). For each identified plant species, we obtained the average size of ripe seeds (length and width, in mm) from published studies and web sources (Appendix A). Additional data from photographs that were unavailable in their respective web sources such as location and date were obtained through direct communication with photographers.

### 4.2. Seed Dispersal Distances

When we were able to detect seeds attached that were transported by parrots during our direct field observations, we measured seed dispersal distances using a laser rangefinder (Leica Geovid 10 × 42, range of measurements: 10–1300 m) [29]. To record seed dispersal distances, we only considered the distance between the mother plant where the parrot was feeding and the last position that we could record after the parrot flew. Given that vegetation can hinder the monitoring of parrots while flying, we considered a minimum dispersal distance as a conservative measurement.

## Figures and Tables

**Figure 1 plants-10-00760-f001:**
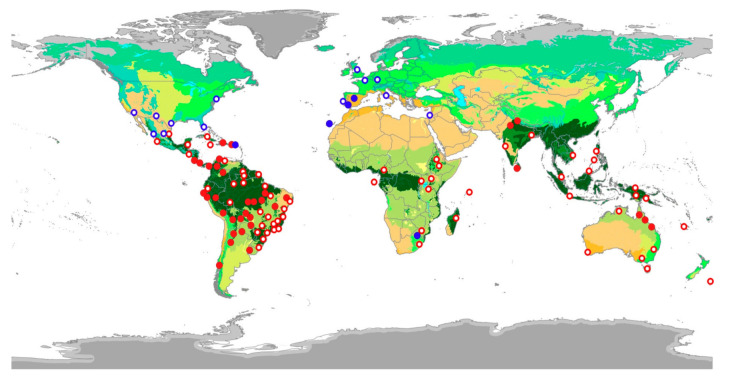
Locations of native (red circles) and non-native parrots (blue circles) recorded carrying attached seeds, with filled circles indicating our own observations and unfilled ones those recorded by wildlife photographers. Colors in the map represent the main terrestrial biomes of the world (https://www.worldwildlife.org/publications/ecoregions-map, accessed on 11 April 2021).

**Figure 2 plants-10-00760-f002:**
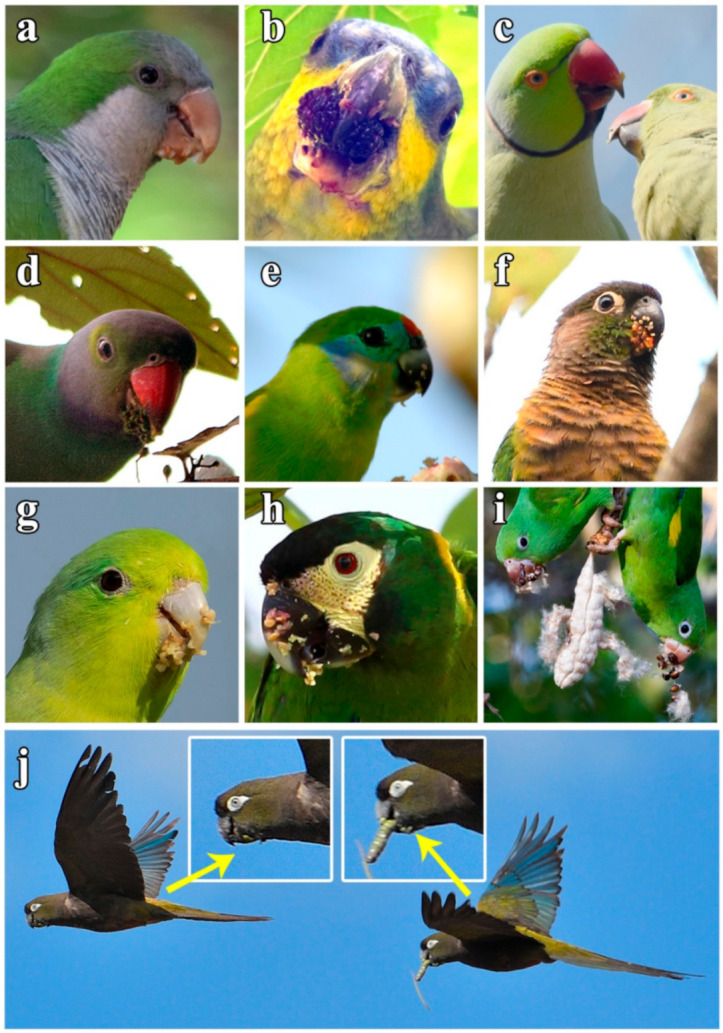
Some examples of seeds adhered to parrots beaks while feeding on fruits: (**a**) monk parakeet *Myiopsitta monachus*, Spain, (**b**) orange-winged amazon *Amazona amazonica*, Spain, (**c**) rose-ringed parakeet *Psittacula krameri*, South Africa, (**d**) emerald-collared parakeet *Psittacula calthrapae*, Sri Lanka, (**e**) double-eyed fig parrot *Cyclopsitta diophthalma*, Australia, (**f**) green-checked parakeet *Pyrrhura molinae*, Argentina, (**g**) blue-winged parrotlet *Forpus xanthopterygius*, Bolivia, (**h**) yellow-collared macaw *Primolius auricollis*, Bolivia, (**i**) yellow-chevroned parakeet *Brotogeris chiriri*, Brazil, and (**j**) burrowing parrots *Cyanoliseus patagonus*, Argentina, simultaneously dispersing seeds of *Prosopis nigra* by epizoochory and estomatochory. Photos (**a**–**c**) are from non-native parrot populations. Photos: D. Hernández-Brito (**a**–**d**), J.L. Tella (**e**–**j**), C. Raffel (**g**), J. Widmer (**h**), and F. Rage (**i**).

**Figure 3 plants-10-00760-f003:**
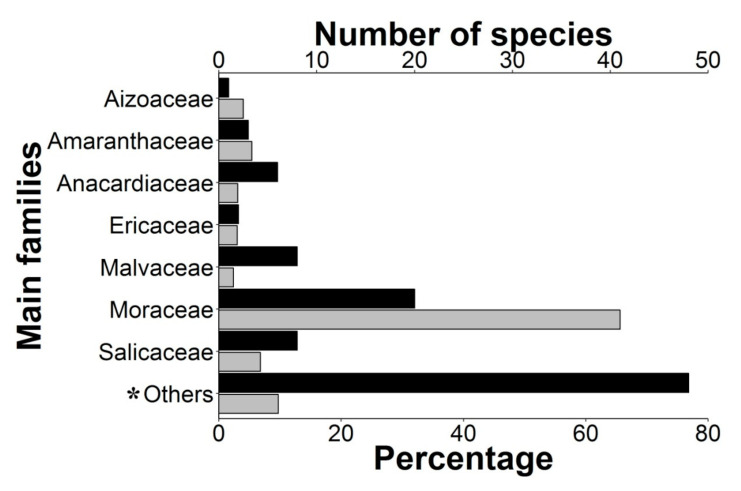
Percentage of events (grey bars) and the number of plant species (black bars) within the main plant families. * Category “Others” includes the plant families with a lower percentage of events (<2.4%): Apocynaceae, Bignoniaceae, Bursareceae, Cactaceae, Cannabaceae, Combretaceae, Compositae, Cupressaceae, Dilleniaceae, Elaeocarpaceae, Euphorbiaceae, Gentianaceae, Lamiaceae, Leguminosae, Malpighiaceae, Melastomataceae, Meliaceae, Monimiaceae, Myrtaceae, Phytolaccaceae, Poaceae, Primulaceae, Rosaceae, Rubiaceae, Sapindaceae, Solanaceae, Urticaceae, and Verbenaceae.

**Figure 4 plants-10-00760-f004:**
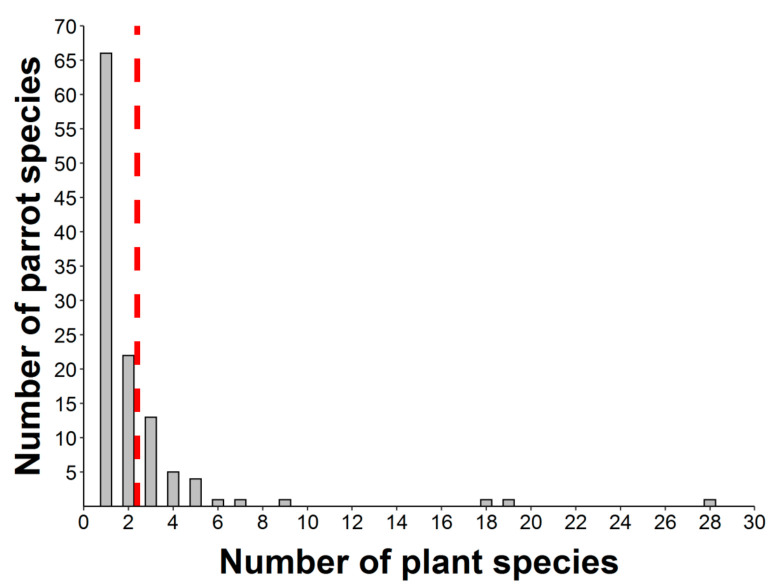
Distribution of the number of epizoochorous interactions between different parrot species and their food plants. Red dashed line shows the mean value of the number of parrot–plant interactions (N = 276).

**Figure 5 plants-10-00760-f005:**
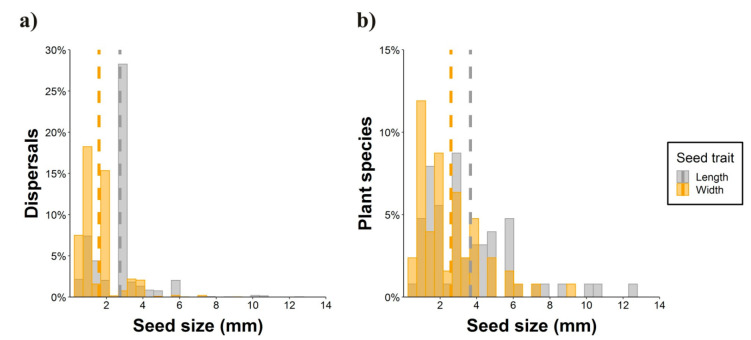
Size (length and width) of the seeds dispersed by parrots through epizoochory considering (**a**) the percentage of seed dispersal events (**b**) and the percentage of dispersed plant species. Dashed lines show the mean values of each seed trait.

**Figure 6 plants-10-00760-f006:**
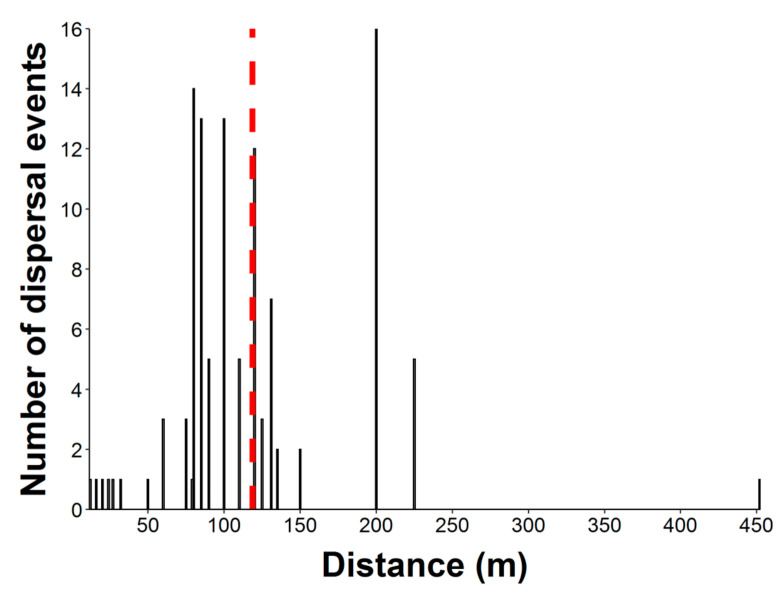
Distribution of seed dispersal distances (N = 112) recorded in 17 parrot species. Red dashed line shows the mean value of seed dispersal distances.

## Data Availability

The raw data supporting the findings of this article has been uploaded as Appendix A. The rest of the raw data are provided in the body of the article.

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
