# Peer review of "Epizoochory in Parrots as an Overlooked Yet Widespread Plant–Animal Mutualism"

_plants, 2021, doi:10.3390/plants10040760_

Round 1
Reviewer 1 Report
This is a very interesting study of how parrots around the world are dispersing seeds through epizoochory. It should alert researchers to similar interactions between other taxa. Your results should also open the door to follow-up experimental and observational studies to determine the fates of seeds dispersed by epizoochory and effects on plant communities. The inclusion of evidence from wildlife photographers adds a valuable component to the results, and it should encourage other researchers to involve citizen scientists in similar or other ways.
My main suggestion is to include definitions of all the technical terms for modes of seed dispersal in the Introduction. It would be helpful to many readers, especially citizen scientists, to define, describe, or compare estomatochory, endozoochory, and anemochory all at once in the Introduction (as you do for epizoochory), or at least as they occur elsewhere in the text. The narrative is generally well-written and clear, but there are instances where the choice of a word or phrase could be improved for readability and conformity to standard style and syntax (please see the following lines).
47: …spreading success of their food plants… >> …dispersal of their food plants…
62: drop >> detachment
67: could >> can; delete finally
68: unfrequent >> infrequent
69: …which is even more unknown… >> …which is even less known [understood]…
84: evidences >> evidence
119, 229: estomatochory >> need to define this term
137: anemochorous >> need to define this term
145: epizootic >> this word generally refers to a disease outbreak among non-human animals, so another word or phrase should be used in its place
224: …plant invasions can be facilitated by native frugivores [45] as well as outcompete native plants for the attention of potential seed dispersers [43,46]. >> …plant invasions can be facilitated by native frugivores [45] and invasive plants can outcompete native plants for the attention of potential seed dispersers [43,46].
239: …will assist to develop effective conservation actions… >> …will assist [facilitate] the development of effective conservation actions…
Author Response
Reviewer 1:
This is a very interesting study of how parrots around the world are dispersing seeds through epizoochory. It should alert researchers to similar interactions between other taxa. Your results should also open the door to follow-up experimental and observational studies to determine the fates of seeds dispersed by epizoochory and effects on plant communities. The inclusion of evidence from wildlife photographers adds a valuable component to the results, and it should encourage other researchers to involve citizen scientists in similar or other ways.
My main suggestion is to include definitions of all the technical terms for modes of seed dispersal in the Introduction. It would be helpful to many readers, especially citizen scientists, to define, describe, or compare estomatochory, endozoochory, and anemochory all at once in the Introduction (as you do for epizoochory), or at least as they occur elsewhere in the text. The narrative is generally well-written and clear, but there are instances where the choice of a word or phrase could be improved for readability and conformity to standard style and syntax (please see the following lines).
Thank you very much for these positive comments, we appreciate very much all the detailed suggestions provided. We have addressed all of them since we feel they are helping us to improve the clarity of our manuscript. Changes can be easily seen in the new version with tracked changes.
47: …spreading success of their food plants… >> …dispersal of their food plants…
Done.
62: drop >> detachment
Done.
67: could >> can; delete finally
Done.
68: unfrequent >> infrequent
Done.
69: …which is even more unknown… >> …which is even less known [understood]…
Done.
84: evidences >> evidence
Done.
119, 229: estomatochory >> need to define this term
We have defined it in the text (lines 269-271).
137: anemochorous >> need to define this term
We have defined it in the text (line 207).
145: epizootic >> this word generally refers to a disease outbreak among non-human animals, so another word or phrase should be used in its place
Sorry for the spelling mistake. We have changed it by "epizoochorous interactions".
224: …plant invasions can be facilitated by native frugivores [45] as well as outcompete native plants for the attention of potential seed dispersers [43,46]. >> …plant invasions can be facilitated by native frugivores [45] and invasive plants can outcompete native plants for the attention of potential seed dispersers [43,46].
Done.
239: …will assist to develop effective conservation actions… >> …will assist [facilitate] the development of effective conservation actions…
Done.
Reviewer 2 Report
In this study, which I enjoyed to read, Hernández-Brito et al. assessed epizoochory in parrots after seeds got attached to the animal’s body while feeding in the respective fruits. By using a combination of field observations and seed identification in photographs from multiple internet sources, they identified 116 parrot species dispersing seeds of at least 96 plant species, resulting in a total of 276 specific pairwise interactions. I think that this is a valuable study showing that seed dispersal can occur via seemingly a widespread, yet largely overlooked and neglected, mechanism.
I have no major criticisms to point out. However, I have a few comments and suggestions to make, which, for clarity, I will describe bellow for the respective text sections:
Line 51: Change “can make any of them remain unknown” to “can make any of them to remain unknown”.
Line 51: Delete “even to the researchers studying them”.
Line 69: Delete “which is even more unknown”. Alternatively, change to “which is less known” or a similar expression.
Lines 98-101: These two sentences can be merged into one sentence, providing the overall number of observations and separately those concerning field surveys and those from photos.
Lines 108-109: How many were identified to species-level and how many only to family-level?
Line 131: “Several parrots species”. Please, provide the number of species.
Lines 133-134: The fact that non-native parrots dispersed mostly exotic species (while native parrots show the opposite pattern) could be discussed deeper. The possible implications of this finding are discussed, but there is no brief discussion on its probable causes. Could it be that, in those sites, exotic plant species produce functionally similar fruits (or are even conspecifics) to those that are native in the original range of the parrots?
Lines 134-138: How many species of fleshy-fruited and anemochorous plants were dispersed?
Lines 138-143: If I’m interpreting correctly, these descriptive statistics, together with the Figure 5, tell me that there could be a relationship between parrot size and dispersed seed size. Why do you think that mean seed size in each dispersal event was smaller than mean seed size of the whole dispersed plant species set? Was there any skew in parrot size towards smaller species that could explain the lower mean seed size in each dispersal event? I don’t see any discussion regarding this result.
Lines 172-173: The fact that epizoochory in parrots is an understudied seed dispersal mode is not a finding of this study (it is what the sentence seems to convey). Instead, it is the reason that prompted this study, right? Please, reformulate the sentence.
Line 196: What are those biases?
Lines 212-215: Does this information regarding seed retention time comes from another study? The source of this information must be provided.
Line 246: Can you provide more details on the methodology (e.g., number of transects per site, transect size, etc.)?
Line 251: It is implied here that, in field surveys, all plant species consumed were identified to species-level. However, after inspecting Table S1, I see that it is not the case. I suggest rephrasing to something like “we identified the plant species consumed whenever possible”.
Author Response
Reviewer 2:
In this study, which I enjoyed to read, Hernández-Brito et al. assessed epizoochory in parrots after seeds got attached to the animal’s body while feeding in the respective fruits. By using a combination of field observations and seed identification in photographs from multiple internet sources, they identified 116 parrot species dispersing seeds of at least 96 plant species, resulting in a total of 276 specific pairwise interactions. I think that this is a valuable study showing that seed dispersal can occur via seemingly a widespread, yet largely overlooked and neglected, mechanism.
I have no major criticisms to point out. However, I have a few comments and suggestions to make, which, for clarity, I will describe bellow for the respective text sections:
Thank you very much for these positive comments and the detailed suggestions provided. We have addressed all of them since we feel they help to improve the clarity of our manuscript. Changes can be easily seen in the provided revised version with tracked changes.
Line 51: Change “can make any of them remain unknown” to “can make any of them to remain unknown”.
Done.
Line 51: Delete “even to the researchers studying them”.
Done.
Line 69: Delete “which is even more unknown”. Alternatively, change to “which is less known” or a similar expression.
We have changed it, also in agreement with reviewer 1.
Lines 98-101: These two sentences can be merged into one sentence, providing the overall number of observations and separately those concerning field surveys and those from photos.
We also agree and thus we have merged both sentences (lines 97-99).
Lines 108-109: How many were identified to species-level and how many only to family-level?
We have added the percentage of identification to the species, genus, and family level of plants (lines 108-110).
Line 131: “Several parrots species”. Please, provide the number of species.
We have added the number of parrot species recorded dispersing more than one plant species (line 141).
Lines 133-134: The fact that non-native parrots dispersed mostly exotic species (while native parrots show the opposite pattern) could be discussed deeper. The possible implications of this finding are discussed, but there is no brief discussion on its probable causes. Could it be that, in those sites, exotic plant species produce functionally similar fruits (or are even conspecifics) to those that are native in the original range of the parrots?
Following the comment of the reviewer, we have extended this point by discussing the potential causes of seed dispersal regarding the origin (exotic/native) of both parrot and plant species (lines 243-249). As we explain in the text, we think these dispersal patterns of exotic plants are influenced by the habitat. Our records of exotic parrot species were mostly obtained in urbanized environments (see details added in Results, lines 145-146), where the diversity of exotic animals and plants is higher than in natural areas (Cadotte et al. 2017; Gaertner et al. 2017). Nonetheless, the consumption of exotic plants by both native and exotic parrots in urban areas can trigger their spread across surrounding natural landscapes through seed dispersal (Gosper et al. 2005; Buckley et al. 2006; Gelmi-Candusso and Hämäläinen 2019).
Regarding the functional similarity of fruits, it is true that some exotic plants in urban environments are in turn native in the original range of the most successful invasive parrot species, such as the rose-ringed and the monk parakeets. For instance, we have observed several fig species (genus Ficus) that are native to the Indian subcontinent, southeastern Asia, and Africa, thus coinciding with the native range of the rose-ringed parakeet. Therefore, we have also added this point in the discussion (lines 255-265).
Lines 134-138: How many species of fleshy-fruited and anemochorous plants were dispersed?
We have added the percentage of plant species that showed each type of seed attachment. Besides, we have incorporated a third- less frequent category ("others"), in which some dry fruits (e.g., capsules and legumes) are included (lines 151-153). In this category, seeds showed an attachment mechanism similar to fleshy fruits, although viscid substances are not from the pulp (e.g., resin and mucilage). Additionally, we have added a new column in Table S1 with this categorization of attachment mechanisms (see Materials and Methods, lines 295-296).
Lines 138-143: If I’m interpreting correctly, these descriptive statistics, together with the Figure 5, tell me that there could be a relationship between parrot size and dispersed seed size. Why do you think that mean seed size in each dispersal event was smaller than mean seed size of the whole dispersed plant species set? Was there any skew in parrot size towards smaller species that could explain the lower mean seed size in each dispersal event? I don’t see any discussion regarding this result.
We understand the point of the reviewer, but we do not think there is a relationship between the sizes of parrots and seeds. On the one hand, we think that the affinity to disperse the smallest seeds is because the attachment of tiny seeds is easier than for bigger ones. On the other hand, near 90% of our records showed seed sizes smaller than or equals to the mean seed size of the plant species set (see Results, lines 159-163), for both measurements (length and width). We have clarified this point in the discussion (lines 232-235). Nonetheless, in this paper, we just aim to show a widespread, yet neglected epizoochory mutualism that we feel is of interest for researchers interested in plant-animal interactions. We are continuing recording data and planning a further paper where we will examine in detail the relationships between fruit and seed size of plants dispersed by different mechanisms by parrots (epizoochory, estomatochory, and endozoochory) and the traits of the parrot species involved.
Lines 172-173: The fact that epizoochory in parrots is an understudied seed dispersal mode is not a finding of this study (it is what the sentence seems to convey). Instead, it is the reason that prompted this study, right? Please, reformulate the sentence.
We agree and have reworded this sentence to clarify it (line 192).
Line 196: What are those biases?
The use of the word "biases" is incorrect in this sentence. We were referring to observational limitations when recording epizoochory from other frugivorous and granivorous birds, as we mentioned in the case of parrots in the Introduction (lines 81-84). We have reworded this sentence for clarity (lines 215-217).
Lines 212-215: Does this information regarding seed retention time comes from another study? The source of this information must be provided.
The observation of seed retention comes from our own observations during the study, so we have clarified this point in the sentence (line 238).
Line 246: Can you provide more details on the methodology (e.g., number of transects per site, transect size, etc.)?
We have included more details about the number of transects and total km covered in the new version of the manuscript (lines 288-289), although this information was only available for roadside surveys as we did not record the effort done on walking transects when looking for foraging parrots. More information about road transects and their validity to monitor wild parrot populations is provided in [61].
Line 251: It is implied here that, in field surveys, all plant species consumed were identified to species-level. However, after inspecting Table S1, I see that it is not the case. I suggest rephrasing to something like “we identified the plant species consumed whenever possible”.
We also agree that it is incorrect, so we have rephrased this sentence (lines 296-297).
Reviewer 3 Report
The authors need to explain why they focused on parrots since the phenomenon they report should also occur in many other frugivorous bird group. The argument that the fruit manipulation abilities of parrots may facilitate high rates of seed attachment to the beak and surrounding plumage (L. 198-199) is not clear. Other frugivorous birds (e.g., tanagers) also mandibulate fruits frequently and should be as likely as parrots to carry adhered seeds in the way reported in this study.
Authors report maximum seed length and width of 29 mm and 9.25 mm respectively, but such extreme values do not appear on Fig. 5a. It would be nice to inform in the Results section which plant species were these.
Author Response
Reviewer 3:
The authors need to explain why they focused on parrots since the phenomenon they report should also occur in many other frugivorous bird group. The argument that the fruit manipulation abilities of parrots may facilitate high rates of seed attachment to the beak and surrounding plumage (L. 198-199) is not clear. Other frugivorous birds (e.g., tanagers) also mandibulate fruits frequently and should be as likely as parrots to carry adhered seeds in the way reported in this study.
We focused on parrots because, as shown in Methods and Discussion, we were studying parrots and their ecological functions for years, and thus we found they are also acting as epizoochorous dispersers. We agree with the point that other frugivorous birds may also play a similar, overlooked role, as we indicate in lines 215-217.
Authors report maximum seed length and width of 29 mm and 9.25 mm respectively, but such extreme values do not appear on Fig. 5a. It would be nice to inform in the Results section which plant species were these.
Thanks for noting us this mistake, the maximum value of length seed is not 29 mm but 12.5 mm, according with the value ranges shown in the graphics (Fig. 5). We have corrected it in the text and we have added the plant species that show maximum and minimum values of seed sizes in Results (lines 153-156).